# ELEMENT2VEC: BUILD CHEMICAL ELEMENT REPRESENTATION FROM TEXT FOR PROPERTY PREDICTION

## ABSTRACT

Accurate property data for chemical elements is crucial for materials design and manufacturing, but many of them are difficult to measure directly due to equipment constraint. While traditional methods use the property of other elements, or related properties for prediction via numerical analyses, they often fail to model complex relationships. After all, not all characteristics can be represented as scalars. Recent efforts have been made to explore advanced AI tools such as language model for property estimation, but still suffer from hallucinations and a lack of interpretability. In this paper, we investigate Element2Vec to effectively represent chemical elements from natural languages to support research in the natural sciences. Given the text parsed from Wikipedia pages, we use language models to generate both a single general-purpose embedding (*Global*) and a set of attribute-highlighted vectors (*Local*). Despite the complicated relationship across elements, the computational challenges also exists because of 1) the discrepancy in text distribution between common descriptions and specialized scientific texts, and 2) the extremely limited data, *i.e.*, with only 118 known elements, data for specific properties is often highly sparse and incomplete. Thus, we also design test-time training method based on self-attention to mitigate the prediction error caused by Vanilla regression clearly. We hope this work could pave the way for advancing AI-driven discovery in materials science.

## 1 INTRODUCTION

Accurate property data of chemical elements, *e.g.*, *atomic radius and conductivity*, are of paramount importance to materials design and manufacturing, *e.g.*, *barrier material deisgn*. Nevertheless, the direct measurement of specific properties for certain elements presents inherent challenges, *e.g.*, *"Lithium-Ion diffusion barrier"* where measuring ionic conductivity of Lithium requires complicated synthesizing for dense ceramic pellets and thin films while the results are highly sensitive to materials defects. As an alternative, numerical analysis that leverages other properties of the element, *e.g.*, *anion size* or analogous properties of related elements, *e.g.*, *Ti, Ge, Zr*, are employed, which may still require profound expert knowledge and fail to model intricate element-wise relationship comprehensively, resulting in limited effectiveness.

With the recent advance of deep learning, the graph neural networks have emerged as powerful tools to implicitly learn element correlation for property prediction (LeCun et al., 2015; Shi et al., 2024; Omee et al., 2022; Pyzer-Knapp et al., 2022; Handoko & Made, 2025; Hu & Tian, 2023; Li, 2025), but may always require the data of properties used in input be complete. In contrast, the data in materials science is of inherent and severe sparsity.Then, the large language models (LLMs) are finetuned for automatic chemical knowledge extraction from literature, followed by encoding in pre-defined formats (Van Herck et al., 2025; Xie et al., 2024; Jacobs et al., 2024), and direct material property prediction from chemical composition (Madika et al., 2025; Choudhary et al., 2022). Nevertheless, both approaches still rely on one-hot or handcrafted descriptors for elements, hindering the generalizability. Furthermore, issues like hallucination (Huang et al., 2025) and data distribution shift can compromise the reliability and utility of LLM further.

In this work, we propose to employ LLM as a embedding extractor, and construct vector representation of chemical elements from free-form textual descriptions. For one thing, the materials are multifaceted entities with the description requiring a synthesis of physical, chemical, and structural

Figure 1: **Encode Element Representation from Text based on inspiration from Image Representation.** (Left) Conventional image representation can be either a global embedding extracted from the whole image, or a set of local embeddings extracted from sub-region that focus on several key components such as tail and head. (Right) Analogously, given the tramendous amount of text in element description and its diverse application and aspects, we can also extract a single *Global* embedding from all text or following the defintion in (Cardarelli, 2008) to extract *Local* embeddings highlighting on different element attributes like atomic, chemical, thermal and optical.

attributes (Willatt et al., 2019). Then, the many properties, *e.g.*, *phase behavior such as gas and liquid*, as well as material design process, cannot be intuitively represented as mathematical descriptors. For another thing, the strong representation encoded with multifaceted characteristics may enable application of simple algorithm for reliable and explainable prediction (Jordan & Mitchell, 2015). Thus, for each element, by integrating diverse information into a unified, context-rich representation (Jin et al., 2025), we may enable broader downstream applications.

Specifically, we propose Element2Vec that leverages the LLM to extract embeddings from text parsed from Wikipedia webpage and these embeddings can be directly compared in the same representation space. As illustrated in Fig. 1 (Left), motivated by the image encoding techniques that produce both a single embedding to describe entire image (globally) and a set of localized embeddings focusing on specific sub-regions, for each element, in addition to a *Global* embedding from the entire Wikipedia page to capture holistic information, we also design the streamline to extract a set of *Local* embeddings, where each is tailored to a specific attribute category, *e.g.*, chemical and atomic, following the taxonomy defined in (Cardarelli, 2008). Without loss of generality, we assess properties framed as both classification and regression tasks. In particular for regression under high data sparsity, we propose a test-time training method based on self-attention that recasts prediction as an imputation problem to enhance precision. With the contribution summarized as follows, we hope that our efforts could open new avenues for accelerating materials design (Jiang et al., 2025):

- We investigate the representations of chemical elements from text by leveraging the capabilities of LLM. Moving beyond numerical data, we believe that integrating diverse information formats into a unified representation is essential for improving downstream tasks like property estimation.

- We investigate Element2Vec, a training-free framework that generates and compares a *Global* embedding distilled from the entire text corpus, *i.e.*, Wikipedia, and *Local* embeddings that highlights specific attributes.

- Property estimation involves both problems in format of classification and regression. For regression under high data sparsity, we propose a test-time training method based on self-attention and formulate the problem as imputation.

## 2 RELATED WORK

**AI for materials discovery.** Deep learning graph networks such as CGCNN (LeCun et al., 2015) and MEGNet (Omee et al., 2022) have been applied for property prediction (Shi et al., 2024), while bond angles are further incorporated to improve predictions (Kaundinya et al., 2022; Choudhary & DeCost, 2021). Unsupervised natural language processing technique have als o uncovered chemical knowledge (Tshitoyan et al., 2019). With the recent success of language model, MatSciB-ERT (Gupta et al., 2022) is designed and tailored to materials text for information extraction and

LLMs like GPT-3 (Brown et al., 2020) is fine-tuned on chemistry tasks can predict molecular and materials properties competitively in low-data regimes (Van Herck et al., 2025; Xie et al., 2024; Jacobs et al., 2024).

**Chemical elements embedding.** Early work used physical properties or compositional data to project elements into vector spaces that reflect periodic trends. For instance, methods based on self-organizing maps (Ward et al., 2018) and Magpie features (Lemes & Dal Pino, 2011) successfully grouped elements by similarity, while unsupervised approaches like Atom2Vec (Maaten & Hinton, 2008) and SkipAtom (Antunes et al., 2022) learned embeddings from compound structures that mirrored the periodicity of properties in an unsupervised manner. However, the dimensionality-reduction techniques (Maaten & Hinton, 2008; Jolliffe, 2011) primarily rediscover known relationships from existing data, lacking a mechanism for extrapolation beyond established knowledge. Nevertheless, all of the methods mentioned above depend on fixed or task-specific elemental encodings. Despite prior explorations with word embeddings and materials knowledge graphs showing that language captures periodic trends and structure–property relations (Tshitoyan et al., 2019; Venugopal & Olivetti, 2024; Ye et al., 2024), the graph model still privilege geometric features but ignores language-informed semantics (Yan et al., 2024; Choudhary & DeCost, 2021; Reiser et al., 2022). Following the great success of large language model, recent efforts emphasize text mining, KG construction, or text-guided generation (Jiang et al., 2025; Das et al., 2025). Very recent "universal atomic embeddings" are trained from crystal data rather than textual descriptions (Jin et al., 2025). Still, the gap in constructing element-level embeddings from text still exists, which motivate our Element2Vec approach that learns broadly applicable element embeddings from text and evaluates them in material property prediction.

**Large Language Models (LLM)** have revolutionized text processing, enabling the flexible extraction of structured knowledge from unstructured text (Liu et al., 2023; Brown et al., 2020). Their capabilities in document segmentation, section identification, and intent classification allow for the precise capture of property-bearing statements from scientific literature (Xu et al., 2024; Dagdelen et al., 2024; Chen et al., 2025). esearch in scientific NLP has developed domain-specialized LLMs to extract structured knowledge from text (Ling et al., 2023; Zhang et al., 2024; Gong et al., 2024). This has spurred the development of domain-specialized LLMs, like SciBERT (Gupta et al., 2024; Dagdelen et al., 2024), which demonstrate superior performance in scientific information extraction. Techniques such as prompting further enable complex tasks like classification and rationale generation without task-specific model fine-tuning. In our pipeline, we leverage these advancements to process Wikipedia text: an LLM segments it into attribute-specific units and filters noisy sentences, thus generating clean, structured input for subsequent embedding and prediction tasks.

**Text-embedding models** convert textual descriptions into fixed-size vectors that preserve semantic relationships (Wang et al., 2024). Early word embeddings like word2vec, utilize word co-occurrence to discern word meanings and identify patterns (Mikolov et al., 2013). Nevertheless, they encounter difficulties in comprehending extended sentences and acquiring detailed knowledge about specific entities. Arora et al. (2017) proposed SIF as a simple yet strong baseline for sentence embeddings, and Logeswaran & Lee (2018) introduced QuickThoughts with a contrastive context-prediction objective. However, many text-only pipelines can be trained upon sufficient amount of data and do not necessarily utilize structured knowledge, *e.g.*, Wikipedia2Vec (Yamada et al., 2018), ERNIE (Sun et al., 2019), KEPLER (Wang et al., 2021). Recent large-scale embedding models leverage LLMs for data generation or as backbones (Wang et al., 2023; Li et al., 2023). In this work, we use Gemini Embedding, which is trained atop the Gemini LLM and explicitly designed to retain knowledge within fixed-size vectors (Lee et al., 2025).

# 3 ELEMENT2VEC: CHEMICAL ELEMENTS REPRESENTATION FROM TEXT

For each chemical element, we utilize the text parsed from its Wikipedia webpage to construct the representation and consider both *Global* and *Local* embeddings.

**Element2Vec-Global.** As an intuitive baseline, given the text corpus $\{\mathbf{s}_i\}_{i=1}^{N_e}$ of element $e$ that consists of $N_e$ sentence $\mathbf{s}_i$. The *Global* embedding is defined as a holistic representation of the entire corpus, which can be obtained as $f_{emb}(\{\mathbf{s}_i\}_{i=1}^{N_e})$ where $f_{emb}$ is implemented as Gemini embedding (Lee et al., 2025).

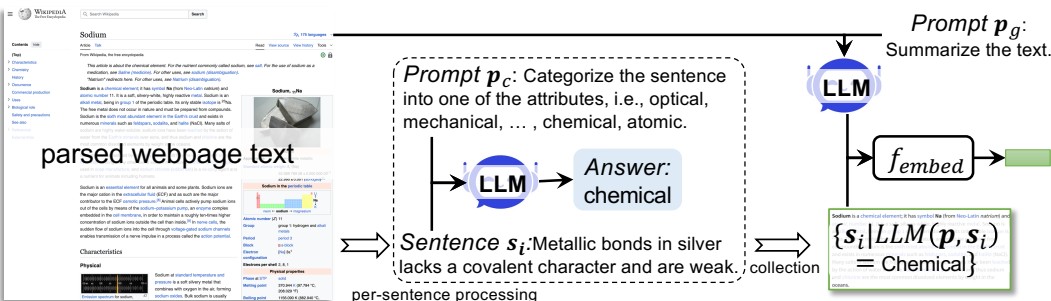

Figure 2: **Element2Vec-Locals Embeddings Extraction.** For each element, given the text parsed from the Wikipedia webpage, we use LLM to label each sentence about the attribute categories (Cardarelli, 2008) independently and summarize the full text to shorten the words via the prompts $\mathbf{p}_c$ and $\mathbf{p}_g$ respectively. For each subset, it is concatenated with the summary and fed into $f_{emb}$ for extraction.

Despite the *Global* embedding aims for a comprehensive representation, a single embedding may struggle to explicitly represent all distinct attributes (Cardarelli, 2008) covered in the content. Meanwhile, the order at attribute level depends on the sentence order, as the original sequential order of sentences preserved in the webpages also vary, the order-sensitive nature of $f_{emb}$ architecture can then introduce noise into the embeddings.

**Element2Vec-Locals** is thus proposed to highlight attribute in the extracted embedding without loosing the global context. For efficiency purpose and without LLM retraining, we extract from a combination from a global summary and a attribute-specific subset, *i.e.*, a filtered collection of sentences, as illustrated in Fig. 2.

In detail, following the concepts in (Cardarelli, 2008), we define eight attribute categories, *i.e.*, "Mechanical (Physical)", "Optical", "Electrical and Magnetic", "Thermal", "Chemical", "Atomic and Radiational", "Applications", and "Abundance". Then, we classify each sentence $\mathbf{s}_i$ using a prompt $\mathbf{p}_c$ that instructs the LLM to assign it to an attribute, *i.e.*, $\text{LLM}(concat(\mathbf{p}_c, \mathbf{s}_i))$. Sentences tagged with the same attribute are automatically grouped into a distinct subset. Finally, by concatenating with a concise global context, $\text{LLM}(concat(\mathbf{p}_s, \{\mathbf{s}_i\}_{i=1}^{N_e}))$, summarized via LLM by the prompt $\mathbf{p}$, we extract the corresponding Gemini embedding (Lee et al., 2025). The justification for including both components is provided by the ablation studies in Sec. 4.1.

## 4 PROPERTY PREDICTION BY ELEMENT2VEC

We first evaluate the characteristics of the *Global* and *Local* embeddings via element classification Secs. 4.1 and 4.2. Subsequently, we apply these embeddings to property estimation framed as a regression problem, introducing and demonstrating the efficacy of test-time training approach Secs. 4.3 and 4.4 even under conditions of high data sparsity.

### 4.1 CLASSIFICATION OF CHEMICAL ELEMENTS

In this paper, we define family as a group of elements in the periodic table that share similar chemical properties, electronic configurations, and reactivity patterns. Specifically, families are defined as the vertical columns (element groups), e.g.: Alkali metals (Group 1: Li, Na, K, . . . ). Family classification is the process of categorizing elements into these families.

We use t-SNE to compare Element2Vec-Global and Element2Vec-Locals representations under identical projection hyperparameters and a fixed random seed (Fig. 3). Fig. 3(a) feeds the entire Wikipedia page to the Gemini embedding model with an added instruction prefix ("prompt") intended to induce attribute-aware encoding. Fig. 3(b) uses an LLM to segment the page into eight predefined attributes and embeds each attribute text without any summary. Fig. 3(c) applies our Element2Vec-Locals method, in which each attribute text is concatenated with the element-level summary before embedding; the inset figure in Fig. 3(c) highlight two elements to show that the

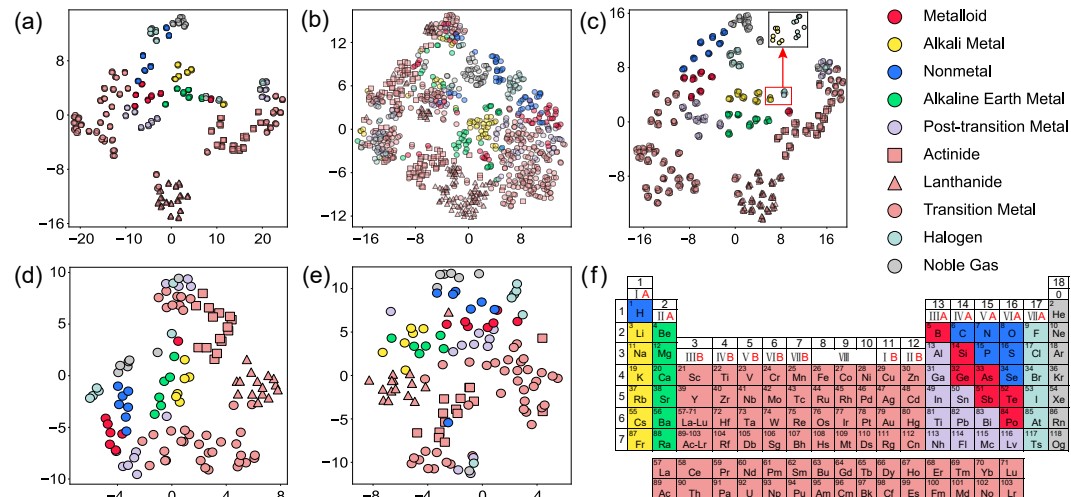

Figure 3: **t-SNE visualization of `element2vec` representations**. (a) Embeddings obtained by sending the entire Wikipedia page with prompt. (b) Local embeddings obtained from 8 attributes without summary. (c) Local embedding, one point per element obtained from the attribute plus its summary. (d) Aggregated embeddings from (c), 8 attribute-level embeddings are combined into a single point per element. (e) Global embedding: one point per element obtained directly. (f) Periodic table with the same color legend as in (a–e).

eight attribute points of the same element form a compact local cluster rather than collapsing to a single point. Fig. 3(d) aggregates the eight attribute embeddings from Fig. 3(c) into one vector per element by concatenation (details in Sec. 3), whereas Fig. 3(e) shows the Element2Vec-Global pipeline where one vector is obtained directly from the full page. Fig. 3(f) provides the periodic table with the same color legend; colors denote chemical families and markers distinguish lanthanides (triangles) and actinides (squares).

Three observations follow. First, in Fig. 3(a) the attribute-level points of a given element nearly coincide, indicating that adding a prompt to the full-page input does not induce meaningful attribute selectivity; the representations behave almost as if the same text were embedded repeatedly, which means that our prompt do not worked in Gemini embedding. Second, in Fig. 3(b) the eight attributes per element scatter broadly and do not form identifiable per-element clusters, suggesting that removing the shared global context makes attribute snippets too heterogeneous to anchor to their source element. Third, Fig. 3(c) yields a markedly more coherent geometry: for nearly all elements, the eight attribute points cluster together while retaining small intra-cluster spread, consistent with the design goal of preserving both shared element identity (via the summary) and attribute-specific variation (via the segmented text).

At the element level, Fig. 3(d) (Element2Vec-Locals) aligns more closely with the periodic families than Fig. 3(e) (Element2Vec-Global). Families such as metalloids, alkali metals, nonmetals, and alkaline earth metals form tighter, cleaner groups in Fig. 3(d), and transition metals—including lanthanides and actinides marked by different symbols—also exhibit improved compactness. Because all panels share identical t-SNE settings, these differences reflect the representations themselves. In summary, summary-augmented Element2Vec-Locals embeddings produce attribute-level micro-clusters anchored on their parent elements and yield Element2Vec-Locals that respect periodic families more faithfully than Element2Vec-Global.

## 4.2 ENTROPY ANALYSIS WITH SUMMARY-AUGMENTED ELEMENT2VEC-LOCALS EMBEDDINGS

Based on Fig. 3(d) and Fig. 3(e), where Element2Vec-Locals yielded tighter intra-element clusters and clearer family-level separation than Element2Vec-Global, we now move from qualitative visualization to a quantitative assessment of prediction decisiveness.

Encoding an entire element page into a single vector can dilute attribute-specific signals, whereas encoding only attribute snippets may lose the global context that relates an element's properties. We therefore adopt a Element2Vec-Locals scheme augmented with an element-level summary: each Wikipedia page is segmented into eight predefined attributes; a concise summary of the full page is produced. For each attributes, the attributes text is concatenated with the summary (front or end), encoded with the Gemini embedding model, and used in downstream tasks. We evaluate three datasets: Element2Vec-Global (global), Element2Vec-Locals with summary prepended (local, front), and Element2Vec-Locals with summary appended (local, end).

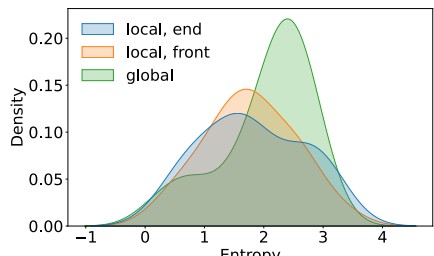

Figure 4: **Entropy distributions for family classification.** Kernel density estimation (KDE) of prediction entropy for the 10-way periodic-family classifier under three embeddings.

To quantify predictive uncertainty for the 10-way family classifier, we compute the Shannon entropy of the softmax posterior for each test element $x$,

$$H(x) = -\sum_{k=1}^{K} p_k(x) \log_2 \big( p_k(x) + 10^{-9} \big), \tag{1}$$

where $K = 10$ and $p_k(x)$ is the predicted probability of class $k$; lower $H(x)$ indicate sharper, less ambiguous posteriors. The constant $10^{-9}$ is to ensure $\log_2 0$ is never evaluated when $p_k(x) = 0$.

The resulting distributions (Fig. 4) show that both `local, front` and `local, end` shift entropy toward lower values compared with `global`, indicating sharper posteriors and reduced ambiguity. Between the two summary placements, `local, end` consistently exhibits a slightly larger mass at lower entropies than `local, front`. A possible interpretation is that appending the summary preserves attribute-specific leading context while still injecting a shared global prior, yielding marginally more decisive predictions. Overall, augmenting attribute segments with a brief summary reduces uncertainty relative to a single global embedding, and placing the summary at the end offers a modest additional gain.

### 4.3 Test-Time Training for Robust Property Value Imputation

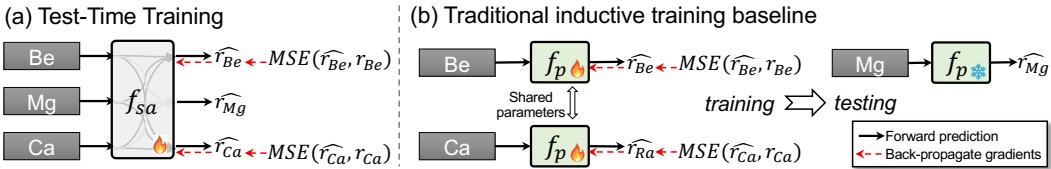

Figure 5: Pipeline comparison between (a) test-time training and (b) standard inductive training.

Different from the family classification problem formulated above where the labels of all elements are given, a key objective for property prediction is to estimate the property values for certain elements which are unknown or hard to measure.

As an intuitive baseline and shown in Fig. 5 (b), the inductive training pipeline can be considered, where a predictor $f_p$, with typical architecture as linear layer or MLP, is first trained on elements $\mathcal{E}_k$ with known property value and directly tested on the other elements $\mathcal{E}_{uk}$ whose property values are missed, where $\mathcal{E}_k$ and $\mathcal{E}_{uk}$ stands for the set of embeddings for the corresponding element set, *i.e.*, $|\mathcal{E}_k \cup \mathcal{E}_{uk}| = 118$ and $\mathcal{E}_k \cap \mathcal{E}_{uk} = \emptyset$. Then, the core assumption is that the model will generalize from the training elements to unseen test elements, which could be nevertheless sub-optimal when the training data is limited. After all, for certain properties, the property data is highly sparse where the values are available for less-than-20% of the total 118 elements comprising the periodic table.

Motivated by the fact that the properties of an element can be influenced by those of others, it is crucial to model inter-element correlations for accurate prediction. Consequently, we propose

to employ a network $f_{sa}$ based on self-attention architecture (Vaswani et al., 2017) and design a corresponding test-time training pipeline as shown in Fig. 5(a) as the training of network parameters and the property prediction of elements in $\mathcal{E}_{uk}$ can be done simultaneously.

In detail, we formulate the $\mathcal{E}_k \cup \mathcal{E}_{uk}$ as input sequence and fed into the self-attention layer, where the corresponding output of each input embedding is projected via a shared linear layer and used as the predicted property value. Then, for the network supervision, the objective is to only minimize the loss, *e.g.*, mean squared error LeCun et al. (2015) of elements in $\mathcal{E}_k$ and no supervision will be applied for elements in $\mathcal{E}_{uk}$.

In this way, different from the conventional inductive training pipeline, where the gradients used for predictor parameter update is only derived from elements $\mathcal{E}_k$ with known property values, the gradients to update the self-attention parameters are also calculated from elements $\mathcal{E}_{uk}$ with unknown property values, which essentially training the network parameters on $\mathcal{E}_{uk}$ in an unsupervised manner and convert the problem of prediction on $\mathcal{E}_{uk}$ to value imputation. Consequently, the model can be directly trained from scratch and then use the model output at the last step as the final prediction, mitigating the performance drop due to scarce data and the distribution shift between $\mathcal{E}_k$ and $\mathcal{E}_{uk}$.

During implementation, as the true property values of elements in the set $\mathcal{E}_{uk}$ are naturally unknown and cannot be used to quantitatively evaluate the model performance, we instead simulate a scenario by holding out certain elements via random selection as a validation set from $\mathcal{E}_k$. Specifically, we partition $\mathcal{E}_k$ into a training set $\mathcal{E}_{tr}$ and a test set $\mathcal{E}_{te}$, such that $\mathcal{E}_k = \mathcal{E}_{tr} \cup \mathcal{E}_{te}$ and $\mathcal{E}_{tr} \cap \mathcal{E}_{te} = \emptyset$. The ratio $|\mathcal{E}_{te}|\,|\mathcal{E}_k|$ is defined as missing rate, reflecting the difficulty of the property estimation.

For material research, since the atomic number, *i.e.*, the indexes of the elements in the periodic table, is a critical factor in property prediction, the position encoding (Vaswani et al., 2017) should have been incorporated into our algorithm as inputs. However, this is unnecessary from our observation and we think the reason is because that the embedding extracted from the source text already encapsulate the corresponding information, thus rendering explicit encoding redundant.

## 4.4 PREDICTION OF ATOMIC PROPERTIES

Following Section 4.2, to avoid confounding representation changes, all subsequent regression experiments therefore use the summary-prepended setting (`local, front`). Fig. 6 turns to a representative scalar property—the van der Waals (vdW) radius—both to motivate the task and to test whether text-derived embeddings carry element-level size information.

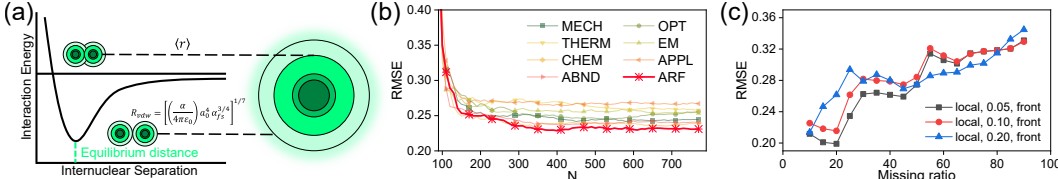

Figure 6: **Evaluation of van der Waals radius prediction via local embeddings.** (a) Schematic of interatomic interaction energy as a function of internuclear separation, showing the definition of the van der Waals radius ($R_{vdW}$). (b) Root mean square error (RMSE) as a function of embedding dimensions $N$ for different local embeddings of elemental attributes, with atomic and radiational features (ARF) highlighted in red. (c) RMSE versus data missing ratio under different ways of constructing the local embeddings.

Fig. 6(a) shows the interaction–distance curve and the $R_{\text{vdW}}$ relation used here, underscoring why $R_{\text{vdW}}$ matter: they enter noncovalent contact a steric accessibility, packing, and force-field parameterization across chemistry, materials, and biomolecular modeling (Zefirov & Zorkii, 1989; Vemparala et al., 2004; Street & Mayo, 1998). At the same time, $R_{\text{vdW}}$ is difficult to determine and not uniquely defined; estimates depend on environment and methodology, often inferred from contact distributions, dispersion models, or polarizability trends rather than measured directly (Mantina et al., 2009; Tkatchenko & Scheffler, 2009; Rahm et al., 2016). This importance–difficulty trade-off motivates evaluating whether our embeddings can predict element-level $R_{\text{vdW}}$.

In Fig. 6(b), we conduct a experiment to validate whether a specific property align with a specific attribute ($R_{\mathrm{vdW}}$-ARF). We first remove elements with any missing attribute text or missing target value, yielding 96 elements. For each of the eight local, front attribute embeddings, we run 10-fold cross-validation and report RMSE on held-out folds. To study feature sufficiency, in every fold a linear regressor is trained after ranking dimensions on the training split and selecting the top $N$ features ($N \in [100, 768]$; selection is performed within the training fold to avoid leakage), and a full-feature linear baseline is also fitted on the same split. As $N$ increases, RMSE decreases and stabilizes; among attributes, Atomic and radiational features attains the lowest error (red curve), consistent with the expectation that $R_{\mathrm{vdW}}$ correlate with atomic size and electronic extent captured by this attribute.

In Fig. 6(c), we vary the summary-to-page ratio in local, front ($0.05, 0.10, 0.20$) and probe robustness under scarce data by sweeping the missing ratio (test fraction) up to $80\%$. For each level we randomize train/test splits five times and average results. Because RMSE can fluctuate with feature count, we aggregate performance using the best-tail average RMSE: identify $N_{\mathrm{best}}$ with the minimum RMSE and average RMSE over $N \in [N_{\mathrm{best}}, N_{\mathrm{max}}]$. As shown in Fig. 6(c), when the missing ratio is below $50\%$, the $0.05$ summary consistently performs best, suggesting that a shorter summary preserves attribute-specific signal while supplying a light global prior. Between $50\%$ and $80\%$, the $0.20$ summary becomes superior, indicating that stronger shared context helps stabilize regression when training evidence is limited and attribute coverage is uneven. Overall, summary-augmented Element2Vec-Locals embeddings support $R_{\mathrm{vdW}}$ prediction, with shorter summaries favored in data-richer regimes and longer summaries acting as a useful prior for sparse data.

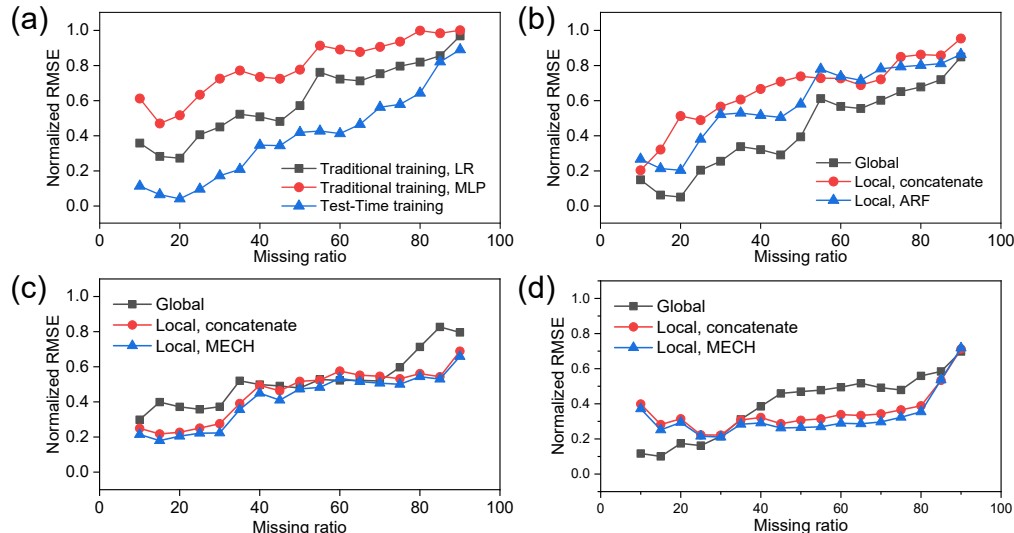

Figure 7: **Model performance comparison against increasingly incomplete dataset.** (a) $R_{vdW}$ predication, for global embedding, the RMSE vs. missing ratio obtained from traditional training and test-time training. (b) $R_{vdW}$ predication task, RMSE vs. missing ratio predicted via linear regression for global, local-concatenated, and local-ARF embeddings (with 0.05 summary ratio putting at the front). (c) Thermal conductivity prediction. (d) Young's modulus prediction.

In Fig. 7(a), we observe that test-time training consistently performs better than the traditional training approaches (both linear regression(LR) and MLP), particularly as the missing ratio increases. This suggests that adapting the model can effectively mitigate the impact of incomplete inputs, leading to substantially lower RMSE values across all missing ratios especially at the higher ones. In Fig. 7(b), we show the trends of the influence of different embedding strategies. Fig. 6(c) has already showed that 0.05 summary putting at the front brings the lowest error among the Element2Vec-Locals. Therefore, we decide to show these Element2Vec-Locals results here to comparison against the Element2Vec-Global. Interestingly, the global embedding generally exhibits the lowest error across most missing ratios, indicating that Element2Vec-Global are inherently more accurate than localized representations for property predictions. These insights point to the combined potential

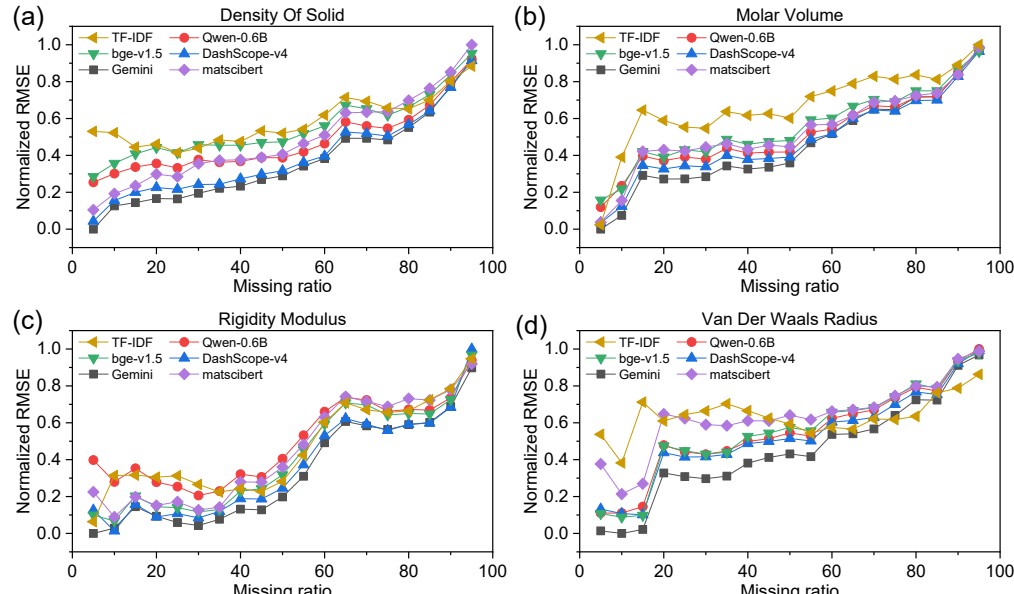

Figure 8: **Robustness evaluation of Element2Vec embeddings with other embedding models.** Normalized RMSE is reported for predicting four representative materials properties: (a) Density of solid, (b) Molar volume, (c) Rigidity modulus, and (d) Van der Waals radius. Performance is compared across five baselines—Qwen-0.6B, DashScope-v4, bge-v1.5, MatSciBERT Gupta et al. (2022), and TF-IDF.

of Element2Vec-Global with test-time training as a promising direction for building reliable models under real-world conditions where data incompleteness is unavoidable. The selected representative elements predictions of $R_{\text{vdW}}$ via Element2Vec-Global with 95% confidence interval are presented on Table 3 in Appendix. To further compare the performance between local embeddings to global embeddings, we further conducted experiments on two additional materials properties. Fig. 7(c) and 7(d) report the normalized RMSE as a function of missing ratio for Young's modulus and thermal conductivity, respectively. For both cases, the local embeddings associated with the corresponding attributes yield lower RMSE than the Global embedding.

To demonstrate a broader suite of materials properties generated through the Element2Vec workflow, we compare the prediction accuracy across several well-established baseline models. As shown in Fig. 8, these include both fundamental atomic properties (e.g., molar volume) and mechanical descriptors (e.g., rigidity modulus). Consistently, Element2Vec maintains good performance even under substantial information removal, which shows the robustness and generalizability of the learned elemental representations.

## 5 CONCLUSION

In this paper, we have proposed `Element2Vec`, an LLM-based framework for embedding chemical elements that turns human experienced knowledge into structured materials information. By leveraging Wikipedia page as a main text resource, we construct both *Global* and *Local* attribute-specific embeddings that capture the correlations among elements. Our experiments demonstrate that these embeddings can recover periodic trends and improve element classification. We also show the stability imputation of missing values through test-time training via attention network, in a series of property prediction tasks. Beyond performance gains, the framework integrate large language models with domain-specific structure to uncover latent knowledge and provide transferable, interpretable representations. Future directions will be addressed to build foundation model representations for more generalizable tasks in materials science such as prediction of material composition and synthesis process. The work paves the way of new materials discovery by turning unstructured scientific knowledge into physics-informed representations.

## ETHICS STATEMENT

This work does not involve human subjects, personally identifiable information, or sensitive user data. Our study is based on publicly available Wikipedia pages of chemical elements, which are open-access resources. No privacy, legal, or security issues arise from the use of these data. Potential concerns about bias are limited to coverage imbalance across different elements: certain elements have more detailed descriptions than others. We have explicitly addressed this issue by designing test-time training strategies that mitigate prediction errors when facing incomplete or sparse data. We believe that this work contributes positively to the materials science and machine learning communities by providing new approaches to property prediction while respecting ethical and research integrity standards.

## REPRODUCIBILITY STATEMENT

We have made significant efforts to ensure the reproducibility of our results. All model architectures, hyperparameters, and evaluation protocols are described in detail in the main text and Appendix. For classification and regression tasks, we report averaged results across multiple random splits to minimize variance. We provide pseudocode of the Element2Vec framework in the main text and Appendix, along with specifications of embedding sizes, summary ratios, and training/testing procedures. To facilitate reproduction, the processed dataset (Wikipedia-derived text and attribute segmentation) and implementation scripts will be released in an anonymous repository during the review period. This includes data preprocessing pipelines, embedding generation scripts, and test-time training code. Together, these resources will allow the community to fully replicate and extend our experiments.

## USE OF LARGE LANGUAGE MODELS (LLMs)

Large Language Models (LLMs) were used in this work solely as a technical component of the proposed method (e.g., sentence classification and summarization within the Element2Vec pipeline). They were not employed in the ideation, experimental design, or writing of the manuscript. All conceptual development, analysis, and interpretation were carried out entirely by the authors. The authors take full responsibility for the accuracy and integrity of the content presented. LLMs are not considered contributors and are not eligible for authorship.

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

# A    APPENDIX

The multilayer perceptron (MLP) used in this study has a three-layer architecture consisting of an input layer, a single hidden layer with 100 neurons using ReLU activation, and an output layer. It is configured for efficient training using the Adam optimizer with very weak L2 regularization (alpha=0.0001). The model employs a maximum of 500 iterations and has early stopping enabled to prevent overfitting by halting training if the validation score stops to improve. The statistics of word count for each local attributes and summary information are shown in the Table 1. The abbreviations of attribute names are summarized in Table 2. The detailed performance of all local embedding datasets are summarized in Fig. 9.

Table 1: **Word count of Element2Vec-Locals dataset with different summary ratio(S/R)**. The more detailed word count of our data is up load in attachment.

| S/R | Elem. | Attr.1 | Attr.2 | Attr.3 | Attr.4 | Attr.5 | Attr.6 | Attr.7 | Attr.8 | Sum. |
|---|---|---|---|---|---|---|---|---|---|---|
| 0.01 | He | 1234 | 212 | 540 | 2202 | 2081 | 731 | 2741 | 3112 | 442 |
| | C | 1463 | 666 | 766 | 1535 | 7619 | 5652 | 4168 | 5677 | 291 |
| | Ca | 414 | 37 | 185 | 363 | 3314 | 1837 | 1524 | 1569 | 244 |
| | Ar | 182 | 656 | 987 | 579 | 4155 | 1823 | 3537 | 1508 | 162 |
| | Br | 260 | 520 | 801 | 1475 | 9146 | 843 | 4411 | 1858 | 345 |
| | Au | 1622 | 1217 | 1642 | 172 | 4782 | 1009 | 6346 | 6019 | 567 |
| | Nb | 1294 | 617 | 2254 | 711 | 3856 | 582 | 2764 | 1086 | 256 |
| | Sm | 613 | 1764 | 1960 | 2565 | 4750 | 3067 | 3468 | 1583 | 216 |
| 0.05 | He | 2519 | 901 | 758 | 1950 | 2644 | 2551 | 3043 | 5736 | 1119 |
| | C | 1790 | 675 | 721 | 1452 | 6642 | 3761 | 4302 | 5691 | 1821 |
| | Ca | 616 | 34 | 652 | 257 | 6012 | 3498 | 4215 | 709 | 1408 |
| | Ar | 59 | 629 | 464 | 404 | 2655 | 1357 | 1560 | 644 | 646 |
| | Br | 114 | 730 | 381 | 1664 | 9318 | 585 | 3561 | 1330 | 1730 |
| | Au | 1425 | 1239 | 1897 | 273 | 5050 | 1885 | 6014 | 5381 | 2835 |
| | Nb | 1245 | 1016 | 1957 | 702 | 3589 | 1107 | 5424 | 718 | 822 |
| | Sm | 672 | 1853 | 2768 | 1299 | 4883 | 3682 | 3482 | 1927 | 1325 |
| 0.1 | He | 1996 | 800 | 1065 | 3663 | 3196 | 5414 | 5150 | 5838 | 2027 |
| | C | 2658 | 746 | 714 | 1365 | 7653 | 6621 | 2518 | 5445 | 2620 |
| | Ca | 793 | 34 | 620 | 478 | 5828 | 2784 | 3622 | 1795 | 2817 |
| | Ar | 59 | 568 | 424 | 432 | 2674 | 1585 | 1467 | 486 | 1143 |
| | Br | 244 | 647 | 703 | 3237 | 10074 | 957 | 4476 | 1525 | 3336 |
| | Au | 1333 | 1061 | 1582 | 49 | 5752 | 1090 | 10126 | 6468 | 4486 |
| | Nb | 1209 | 527 | 2225 | 529 | 4204 | 449 | 4761 | 888 | 1504 |
| | Sm | 613 | 1545 | 2066 | 1554 | 4624 | 4690 | 3826 | 2366 | 2650 |
| 0.2 | He | 1608 | 470 | 802 | 3083 | 2249 | 1971 | 1872 | 3565 | 3383 |
| | C | 1801 | 580 | 624 | 1182 | 7590 | 5409 | 2836 | 6681 | 2444 |
| | Ca | 553 | 87 | 510 | 342 | 7695 | 3992 | 3952 | 1738 | 4369 |
| | Ar | 251 | 471 | 403 | 534 | 2136 | 1735 | 2499 | 633 | 1380 |
| | Br | 629 | 420 | 642 | 2417 | 11770 | 761 | 3657 | 1281 | 5104 |
| | Au | 1950 | 1238 | 2060 | 278 | 4604 | 1060 | 5453 | 5570 | 4110 |
| | Nb | 1051 | 814 | 2273 | 629 | 3732 | 583 | 3908 | 1063 | 3040 |
| | Sm | 487 | 1451 | 2527 | 1053 | 5516 | 4218 | 3775 | 2107 | 2306 |

Table 2: Attribute names and abbreviations

| Full name | Abbr. |
|---|---|
| Mechanical properties (Attr.1) | MECH |
| Optical properties (Attr.2) | OPT |
| Electrical & Magnetic properties (Attr.3) | EM |
| Thermal properties (Attr.4) | THERM |
| Chemical properties (Attr.5) | CHEM |
| Atomic & radiational features (Attr.6) | ARF |
| Applications (Attr.7) | APPL |
| Abundance (Attr.8) | ABND |

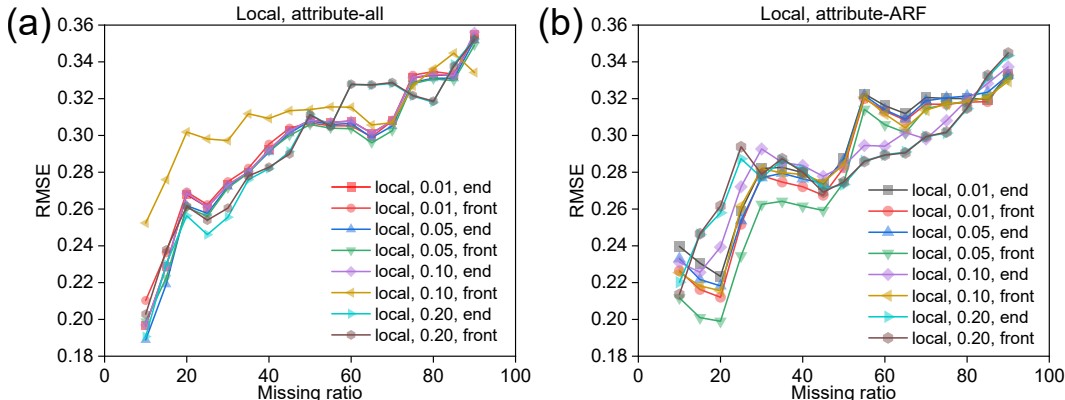

Figure 9: (a) The LR result of all Element2Vec-Locals embedding datasets with full attributes vector. (B) The LR result of all Element2Vec-Locals embedding datasets with only ARF attribute vector.

Table 3: $R_{\mathrm{vdW}}$: **ground truth and predicted values with 95% confidence interval.** The values in the table are in Å; 1 Å $= 10^{-10}$ m.

| Element | True Value | Predicted Value |
|---|---|---|
| He | 1.40 | $1.42 \pm 0.11$ |
| C | 1.70 | $1.65 \pm 0.12$ |
| Ca | 2.31 | $2.27 \pm 0.06$ |
| Ar | 1.88 | $1.78 \pm 0.10$ |
| Br | 1.85 | $1.88 \pm 0.08$ |
| Au | 2.14 | $2.11 \pm 0.06$ |
| Nb | 2.18 | $2.31 \pm 0.04$ |
| Sm | 2.36 | $2.38 \pm 0.06$ |

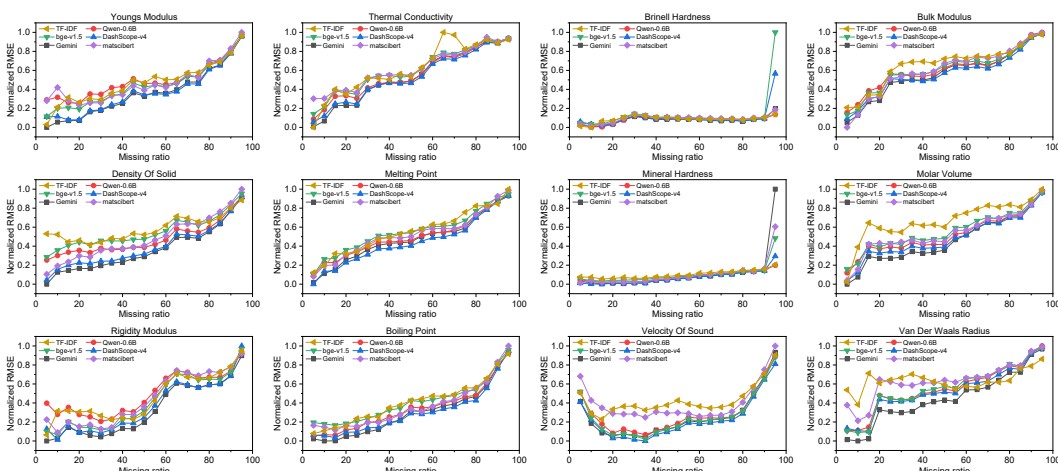

Figure 10: Baseline comparison of different embedding models(Qwen-0.6B, DashScope-v4, bge-v1.5, MatSciBERT, and TF-IDF)in properties prediction.

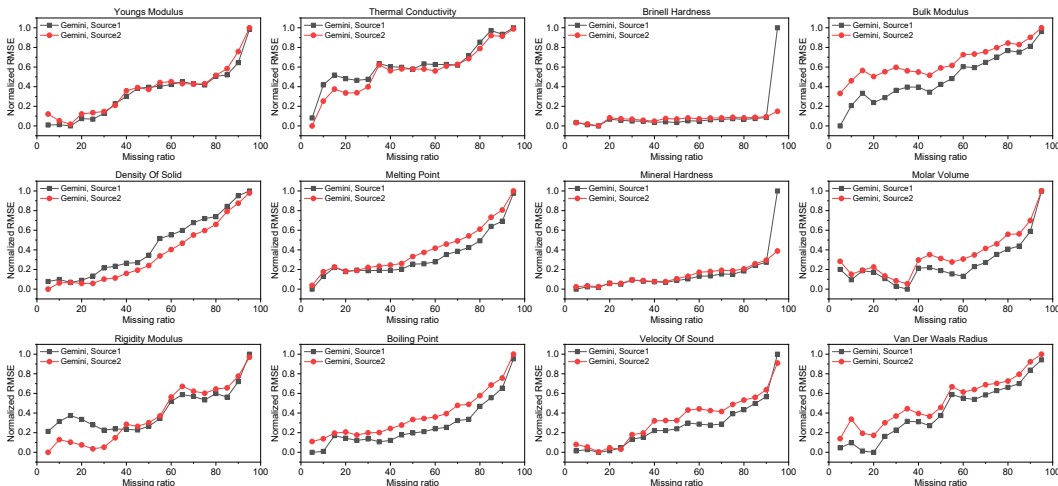

Figure 11: Performance of global embeddings from different sources in prediction task, Source 1 is Wikipedia Page, Source 2 is ElementWiki.

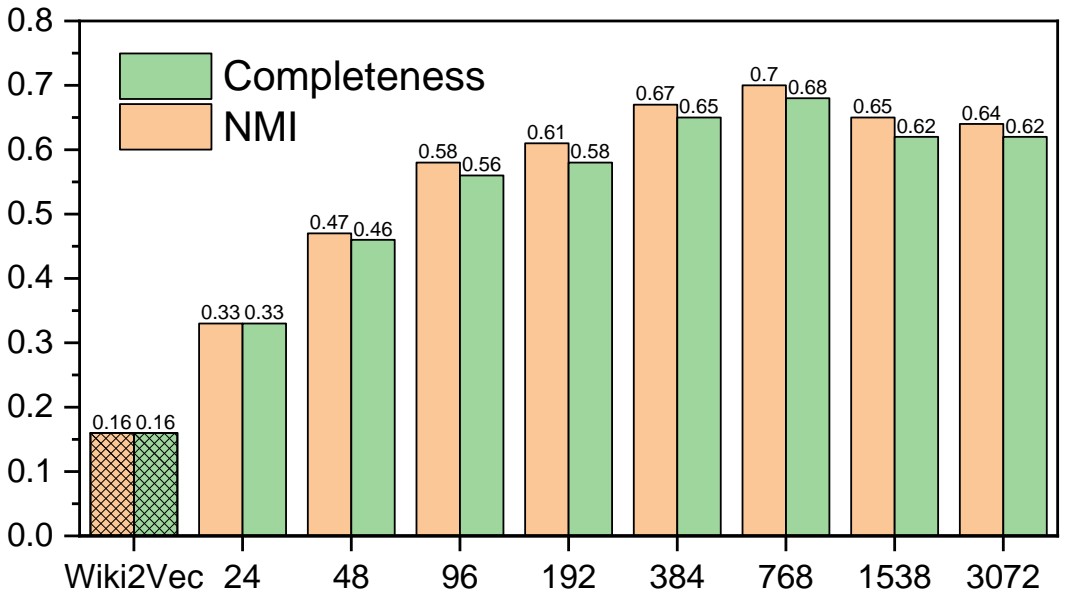

Figure 12: Evaluation of NMI and Completeness scores for naive whole-text embedding (Wikipedia2Vec) and Gemini embedding with different lengths.

Table 4: Sensitivity experiment on test-time training

| Missing ratio/Learning rate | 1.00E-05 | 3.00E-05 | 0.0001 | 0.0003 | 0.001 | DEVSQ |
|---|---|---|---|---|---|---|
| 0.05 | 0.0865 | 0.1148 | 0.0841 | 0.1166 | 0.1953 | 0.008118 |
| 0.1 | 0.093 | 0.2148 | 0.1924 | 0.4125 | 0.249 | 0.054054 |
| 0.15 | 0.2003 | 0.1456 | 0.1647 | 0.2508 | 0.18 | 0.006499 |
| 0.2 | 0.3648 | 0.159 | 0.2443 | 0.1982 | 0.1228 | 0.035178 |
| 0.25 | 0.1605 | 0.2176 | 0.1177 | 0.1338 | 0.3268 | 0.028724 |
| 0.3 | 0.1521 | 0.2139 | 0.2705 | 0.124 | 0.2622 | 0.017 |
| 0.35 | 0.187 | 0.2426 | 0.2018 | 0.1553 | 0.2373 | 0.005261 |
| 0.4 | 0.2779 | 0.296 | 0.2487 | 0.1889 | 0.1463 | 0.015683 |
| 0.45 | 0.2752 | 0.1399 | 0.2776 | 0.2491 | 0.2815 | 0.014369 |
| 0.5 | 0.2404 | 0.2599 | 0.21 | 0.144 | 0.1745 | 0.008941 |
| 0.55 | 0.2334 | 0.2644 | 0.224 | 0.239 | 0.2273 | 0.001029 |
| 0.6 | 0.255 | 0.2869 | 0.2227 | 0.149 | 0.246 | 0.010714 |
| 0.65 | 0.2542 | 0.2689 | 0.2557 | 0.2454 | 0.2263 | 0.00099 |
| 0.7 | 0.3317 | 0.2738 | 0.2848 | 0.2964 | 0.2353 | 0.004905 |
| 0.75 | 0.2687 | 0.2466 | 0.2619 | 0.2713 | 0.2461 | 0.000574 |
| 0.8 | 0.2682 | 0.2677 | 0.2307 | 0.2807 | 0.1914 | 0.005368 |
| 0.85 | 0.3393 | 0.3201 | 0.2249 | 0.2872 | 0.2436 | 0.009492 |
| 0.9 | 0.2993 | 0.316 | 0.3131 | 0.2948 | 0.2492 | 0.002883 |
| 0.95 | 0.3524 | 0.3308 | 0.325 | 0.3238 | 0.3116 | 0.000896 |

