# OpenReview forum: "Element2Vec: Build Chemical Element Representation from Text for Property Prediction"
_ICLR.cc/2026/Conference — Submitted to ICLR 2026_

### Official Review · Reviewer_deg3 · 2025-10-31

**Soundness:** 2
**Presentation:** 2
**Contribution:** 2
**Rating:** 2
**Confidence:** 4

**Summary:**

This paper proposes a framework named Element2Vec, which aims to use LLMs to extract chemical embeddings from the Wikipedia text for material property prediction. The embeddings consist of global embeddings that summarize entire text and local embeddings specific to attribute texts (e.g., mechanical, optical, thermal). To handle the high sparsity in chemical property data, the paper also designs a test-time training strategy based on self-attention, transforming property prediction into an imputation problem. Experimental results show that local embeddings achieve better clustering by element family in t-SNE visualization for classification. In property regression task, the global embeddings combined with the 'test-time training' strategy perform best under high data missing ratios.

**Strengths:**

1.	This paper presents an interesting perspective, using large language models to extract embeddings from text for predicting chemical properties, which can capture richer contextual knowledge missing from traditional databases.

2.	This paper proposes a Test-Time Training strategy, which treats all elements as a whole for 'imputation' prediction. According to the experiments in the paper, this method outperforms traditional inductive training methods at all data missing ratios.

**Weaknesses:**

1.	The authors employ LLM to generate chemical representations from natural language text, however, there are several concerns. The paper fails to sufficiently address the issue of data leakage; since Wikipedia may already contain explicit numerical values or strongly correlated descriptions of the properties being predicted, it is unclear whether the model is learning genuine chemical relationships or merely retrieving and regurgitating memorized information. The authors need to provide a rigorous analysis to rule out this possibility.

2.	Furthermore, the justification for choosing the specific Gemini embedding model over other general-purpose large models (like GPT) or domain-specific models (like MatSciBERT or SMILES-BERT) is insufficient. The paper needs to clearly articulate the specific advantages of the chosen method relative to these alternatives.

3.	The paper's introduction to the global and local embedding generation methods lacks the necessary detail. The process of how text is segmented, summarized, and fed into the model to generate local embeddings is not clearly explained. Concurrently, the paper fails to provide a convincing scientific justification for the necessity of local embeddings. Although the authors hypothesize that local embeddings can highlight specific attributes, the empirical evidence provided appears contradictory, for instance, the results in Figure 7(b) show that the global embedding generally has a lower RMSE in property prediction than the local embeddings, which weakens the motivation for adopting the more complex local embedding method. This work resembles a simple text-processing workflow applied to the chemical domain rather than a substantial new contribution to chemistry or materials science.

4.	The paper lacks modeling of the relationships between local representations. The authors generate independent vectors for attributes (e.g. atomic, chemical, and thermal), but in chemistry, these properties are deeply correlated and interconnected. The current method appears to treat them as mutually independent.

5.	The paper does not provide sufficient detail regarding the dataset used for embedding generation and property prediction. Although Wikipedia is mentioned as the source, the authors need to clearly specify the data collection and cleaning procedures, as well as comprehensive statistics for the final corpus (e.g., average document length, vocabulary size, etc.).

6.	The paper's experimental evaluation is weak due to a lack of adequate baseline comparisons. The authors primarily compare different variations of their own method. A benchmark against non-text-based methods, such as GNNs, is required.

**Questions:**

Please refer to the weaknesses.

---

> ### Author Response · Authors · 2025-11-20
>
> >Weaknesses1 Data Leakage
>
> We thank the reviewer for raising this concern. In the revision we clarify the scraping pipeline. The crawler discards the entire right-hand infobox (Fig.1, Wikipedia webpage) and all tables and keeps only the main prose, so explicit numeric entries (e.g., vdW radius, thermal conductivity) never enter the embeddings; regression labels come from external databases. For vdW radius, Young’s modulus and thermal conductivity we verified that target numbers do not appear in the retained text, and all models are trained and evaluated on disjoint element splits. The smooth generalization curves in Fig. 7d and Fig. 7c thus reflect regularities learned from text, rather than copying tabulated values.
>
> >Weaknesses2 Different Model Comparison
>
> Our method uses nearly complete Wikipedia articles as textual descriptors for elements. These pages often exceed the context-window limits of many common embedding models.
>  Models such as MatSciBERT or SMILES-focused BERT variants are powerful for particular modalities (e.g., chemical text, SMILES strings), but they also encode modality-specific inductive biases and typically benefit from task-specific fine-tuning. Our goal in this work is to construct broad, general-purpose semantic descriptors of elements from encyclopedic text, not to rely on a specific structural representation.
>
> >Weaknesses3 Details About Embedding Method
>
> We feel sorry for not stating the embedding process clearly. We have stated the embedding process in more detail in the revised paper.
>
> >Weaknesses4 Relationships Between Local Representations
>
> In Figure 3(a) and 3(b), we have shown different method of extracting local embeddings and their classification results. While our method produces separate local embeddings for modularity and interpretability, our final representation *does not* assume independence between attributes.
>
> To explicitly address this concern, we highlight that our approach includes what we call the
> **summary-augmented Element2Vec-Locals embeddings**.
> In addition to the eight attribute-specific embeddings, we incorporate a *global summary embedding* derived from the introductory portion of each element’s Wikipedia page. This global summary captures higher-level, cross-property semantic relationships, including chemical behavior, macroscopic properties, and usage contexts. By concatenating the global summary embedding with the local attribute embeddings, our final representation encodes both:
>
>
> 1.**Cross-attribute correlations:**
>     The global embedding provides a shared latent semantic space that naturally reflects interdependencies among physical, chemical, and thermal properties.
>
> 2.**Property-specific interpretability:**
>     Maintaining distinct local embeddings enables inspection of individual attribute contributions while still benefiting from overarching semantic integration.
>
>
> Thus, the final Element2Vec representation integrates both global and local semantics, ensuring that correlated material properties are not treated as independent. We will revise the manuscript to more clearly articulate this design choice.
>
>
>
> >Weaknesses5 Comprehensive Statistics For The Final Corpus
>
> We have shown the vocabulary data of some elements in table1 in the appendix. The vocabulary size for all 118 element’s Wikipedia page is 17432. Because the vocabulary statistics table is large, we will upload it as an attachment.
>
> >Weaknesses6 Baseline Comparisons
>
> We have shown the baseline model comparisons in Figure 8, we will try GNN model in the next few days.

---

> ### Author Response · Authors · 2025-11-27
>
> Dear Reviewer,
>
> I hope this message finds you well. As the discussion period is nearing its end with less than five days remaining, I wanted to ensure we have addressed all your concerns satisfactorily. If there are any additional points or feedback you’d like us to consider, please let us know. Your insights are invaluable to us, and we're eager to address any remaining issues to improve our work.
>
> Thank you for your time and effort in reviewing our paper.

---

### Official Review · Reviewer_EgL7 · 2025-11-01

**Soundness:** 1
**Presentation:** 2
**Contribution:** 1
**Rating:** 2
**Confidence:** 3

**Summary:**

To address the critical challenge of property prediction for chemical elements, this paper employs a Large Language Model (LLM) as an encoding module to extract knowledge from text in the form of embeddings. The proposed Element2Vec framework leverages an LLM to construct vector representations of chemical elements derived from unstructured text sources, specifically Wikipedia pages. The key contribution lies in generating two types of embeddings that capture information at different levels. The global embeddings utilize the entire document or page of an element as input to capture holistic information, while the local embeddings are learned from text grouped under specific attributes. Overall, this idea of using LLM to encode chemical text has been explored in several related works which limit the contribution of this paper.

**Strengths:**

1. The proposed Element2Vec provides an effective way for translating human-experienced, qualitative knowledge like Wikipedia into numerical representations that are machine-readable.

2. The proposed method utilizes both Global and Local (attribute-specific). The local embeddings include the information of specific characteristics (e.g., optical vs. thermal properties), which is vital for materials design and scientific analysis.  The global embeddings capture more holistic knowledge.

3. The training-free framework is another strong aspect. By relying on pre-trained LLMs as feature extractors and content classifiers, the embedding generation pipeline does not require additional training. This makes it straightforward to apply to new elements or attributes without extensive retraining, thereby facilitating faster research and experimentation.

**Weaknesses:**

1. The main limitation of this work is that Element2Vec relies entirely on Wikipedia pages. Consequently, the quality, depth, and neutrality of the generated embeddings depend heavily on the completeness and accuracy of this data source. If a particular element’s Wikipedia page is sparse, outdated, or biased, the resulting embedding may be inaccurate. A potential improvement would be to incorporate additional sources of domain-specific knowledge, such as scientific publications or chemical databases, to enhance representation quality.

2. Wikipedia is a general data source, and most modern LLMs have already been trained on it during pretraining. Therefore, a more appropriate baseline would be to compare this approach against recent, powerful models applied directly to property prediction tasks. Additionally, several models have been fine-tuned for chemistry-related tasks, and including such comparisons would strengthen the paper. Overall, while the application of LLMs to chemical property prediction has been investigated in previous studies, this work would benefit from a clearer demonstration of its unique technical contributions and distinctions from existing approaches.

**Questions:**

Please check the weaknesses.

---

> ### Author Response · Authors · 2025-11-22
> **Response**
>
> > Generalization of Element2Vec
> We clarify that the framework is fully flexible and can incorporate additional textual sources—such as ElementWiki, textbooks, database descriptions, or curated domain summaries, to complete missing attributes when available. In the main paper, we acknowledge that we only studied the Wikipedia pages as the main text source. However, We have also demonstrated that our workflow can process alternative text sources to generate embeddings, indicating that the approach is not restricted to Wikipedia and can readily integrate more structured or domain-specific text sources. We clarified this flexibility in the revised manuscript (See Figure. 11 in the SI: the performance of different text sources in prediction tasks)
>
> > Comparison with Baseline
> The primary contribution of our work is a robust and low-cost framework for processing textual data to predict material properties.
> - **Efficiency**: Finetuning models in chemical data is notoriously expensive and time-consuming, requiring massive and well-curated datasets. In contrast, we propose to directly extract features from web-crawled texts, which clearly reduce the cost in both data curation and computation.
> - **Robustness**: Model finetuning suffers from catastrophic forgetting, i.e., a loss of their original general capabilities, and may lead to "hallucinations" where they generate implausible predictions. In contrast, our evaluation demonstrates that LLM can extract meaningful features from low-quality text for robust property prediction through test-time training. From Figure 8 in the revised version, our method also outperforms the baselines.

---

> ### Author Response · Authors · 2025-11-27
>
> Dear Reviewer,
>
> I hope this message finds you well. As the discussion period is nearing its end with less than five days remaining, I wanted to ensure we have addressed all your concerns satisfactorily. If there are any additional points or feedback you’d like us to consider, please let us know. Your insights are invaluable to us, and we're eager to address any remaining issues to improve our work.
>
> Thank you for your time and effort in reviewing our paper.

---

### Official Review · Reviewer_xsfF · 2025-11-01

**Soundness:** 2
**Presentation:** 1
**Contribution:** 2
**Rating:** 4
**Confidence:** 2

**Summary:**

This paper is concerned about learning representations for chemical elements (e.g., atoms) from text data such as Wikipedia webpages. The proposed representation consists of `Element2Vec-Global` and `Element2Vec-Locals`; the former is a representation of the whole relevant text data, while the latter is a collection of representations, each of which is obtained by prompting an LLM to focus on specific (pre-defined) attribute (details are in Figure 2).
Since the number of all elements is limited, the authors also propose a test-time training approach for prediction, instead of the standard supervised learning (Section 4.3 ).

The benefit of the proposed representations has been validated from several perspectives.
Section 4.1 visually examines the validity of the proposed local representations, as compared against several different representations.
Section 4.2 quantitatively examines the benefit of the local representation.
Section 4.4 examines the effectiveness of the proposed embedding and test-time training method.

**Strengths:**

- Learning a representation from text data is an interesting way of utilizing LLMs.
- It is insightful that the authors have shown several ways to define local embeddings and have explained why the proposed embedding is selected among others.

**Weaknesses:**

One of the major concerns is the empirical validation. As far as I am aware of, a quantitative validation is done on a task to predict the van der Waals radius of an element, without any existing methods to be compared. Since I am not an expert in materials science, I failed to understand the importance of the prediction task, and thus, I thought the experiment was rather a toy task rather than a real-world problem. In addition, without performance comparison with other existing methods, it is difficult to understand whether or not the proposed representation is useful or not in real applications.

**Questions:**

- I would like to ask the authors to clarify how $p_k(x)$ is computed in Section 4.2.
- In Section 4,4, the authors state that "$R_{\mathrm{vdW}}$ is difficult to determine and not uniquely defined", and I'm curious about how the authors determine the ground truth labels.
- I would like the authors to clarify the relevance of the van der Waals radius prediction task to real-world problems.

---

> ### Author Response · Authors · 2025-11-20
>
> >“One of the major concerns is the empirical validation. As far as I am aware of, a quantitative validation is done on a task to predict the van der Waals radius of an element, without any existing methods to be compared. Since I am not an expert in materials science, I failed to understand the importance of the prediction task, and thus, I thought the experiment was rather a toy task rather than a real-world problem. ”
>
>
> The van der Waals radius is only one illustrative example of the properties we examine. Our analysis extends to **multiple additional materials properties** derived using the Element2Vec workflow, for which we also provide quantitative comparisons against several baseline models. (refer to the new Figure 8!)
>
> > “In addition, without performance comparison with other existing methods, it is difficult to understand whether or not the proposed representation is useful or not in real applications. ” This point was also mentioned in Question 3: “I would like the authors to clarify the relevance of the van der Waals radius prediction task to real-world problems.”
>
>
> The van der Waals radius is a fundamental geometric parameter that describes the effective size of an atom when it participates in non-bonded interactions. In materials science, the van der Waals radius plays a crucial role in describing atomic packing, surface interactions, and the structural organization of molecular and layered solids. Accurate knowledge of these radii is essential for understanding intermolecular spacing, adsorption phenomena, and steric effects, all of which significantly influence the thermal, mechanical, and transport properties of materials. The concept is particularly important for modeling soft materials, organic crystals, nanomaterials, and two-dimensional systems, where non-bonded interactions govern stability and morphology. In computational materials science, van der Waals radii are integral to force-field parameterization, collision detection, and molecular dynamics simulations, where they help define excluded-volume interactions and interatomic potentials.
>
>
>
>
>
>
>
> > Question 1: Clarification on $p_k(x)$ computation in Section 4.2.
>
>
> The probability $p_k(x)$ is obtained by feeding elements' embeddings into an XGBoost classifier. The model first generates raw score predictions (logits) for each of the 10 elemental groups. The softmax function is then applied to convert these logits into a normalized probability distribution $p(x)$, where each $p_k(x)$ represents the model's confidence that the sample belongs to group $k$, and all values sum to 1. The highest probability, $\arg\max_k p_k(x)$, is taken as the final value for this element used in computing Shannon entropy.
>
>
>
>
> >
> “Question 2: In Section 4,4, the authors state that "$R_{vdW}$ is difficult to determine and not uniquely defined", and I'm curious about how the authors determine the ground truth labels.”
>
>
> These properties are often challenging to determine experimentally, yet several research groups have curated high-quality materials datasets that provide reliable reference values. In particular, we use property data from the Materials Project as ground-truth labels.(Accelerated data-driven materials science with the Materials Project, Nature Materials 24, 1522-1532 (2025))

---

> ### Author Response · Authors · 2025-11-27
>
> Dear Reviewer,
>
> I hope this message finds you well. As the discussion period is nearing its end with less than five days remaining, I wanted to ensure we have addressed all your concerns satisfactorily. If there are any additional points or feedback you’d like us to consider, please let us know. Your insights are invaluable to us, and we're eager to address any remaining issues to improve our work.
>
> Thank you for your time and effort in reviewing our paper.

---

### Official Review · Reviewer_9G3u · 2025-11-01

**Soundness:** 1
**Presentation:** 2
**Contribution:** 3
**Rating:** 2
**Confidence:** 4

**Summary:**

This paper proposes Element2Vec, a framework for representing chemical elements as vectors by leveraging textual descriptions. The authors use Wikipedia pages for each of the 118 chemical elements as the data source. An LLM-based pipeline is employed to produce two types of embeddings for each element: a single Global embedding capturing the overall content of the element’s page, and multiple Local embeddings that are attribute-specific. To obtain the Local embeddings, the approach first uses an LLM to classify each sentence of an element’s page into one of eight predefined attribute categories (e.g. Atomic, Chemical, Thermal, etc.), and then aggregates the text of each category (with a brief summary of the whole page) to generate an embedding for that attribute . The goal is that these Global and Local embeddings encapsulate meaningful chemical knowledge extracted from text, which can then be used for downstream tasks such as classifying an element’s periodic-table family and predicting various material properties

**Strengths:**

The paper's key strengths lie in its innovative cross-domain approach that bridges NLP and materials science by leveraging large language models to extract knowledge from scientific text, moving beyond traditional hand-designed features. The introduction of attribute-aware embeddings significantly enhances interpretability by producing multiple vectors for each element corresponding to human-understandable categories (mechanical, thermal, chemical properties), which demonstrably organize the latent space in ways that respect known scientific classifications like periodic families. The proposed test-time training method effectively addresses sparse data challenges, showing substantial performance improvements over conventional baselines especially when 50-80% of data is missing, by cleverly allowing unlabeled instances to influence the model during inference. The work is supported by thorough empirical evaluation including comprehensive ablation studies (examining summary length effects, attribute contributions, and embedding dimension overlap) that reveal meaningful insights such as the model's ability to rediscover periodic families from text alone and capture real scientific relationships like the shared features between melting and boiling points. This combination of strong performance, interpretability through attribute-specific analysis, and alignment with domain knowledge makes the approach both scientifically valuable and trustworthy for deployment in materials research.

**Weaknesses:**

The approach suffers from fundamental limitations in its dependence on Wikipedia as a single, uneven source of truth, with sparse coverage forcing the exclusion of 22 elements from certain analyses. More critically, the heavy reliance on LLM-based sentence classification and summarization lacks any validation or accuracy assessment. The paper provides no evidence that the LLM correctly categorizes sentences into attribute buckets or avoids hallucination during summarization, despite these steps being central to the embedding process. This unverified pipeline could propagate errors throughout the embeddings, yet the authors offer no robustness analysis or manual verification of these critical automated decisions.
The experimental evaluation lacks essential baseline comparisons that would contextualize the method's performance. No comparisons are provided against simple alternatives like one-hot encodings, manual feature sets, or naive whole-text embeddings without attribute segmentation. Furthermore, the results reveal a surprising weakness: the sophisticated Local attribute embeddings actually underperform the simpler Global single-vector approach in property prediction tasks, with the authors themselves acknowledging that "global embedding generally exhibits the lowest error." This undermines a core contribution of the paper and suggests the attribute segmentation may lose important holistic information rather than enhance it.
The work's practical applicability is limited by its narrow scope (118 elements only, with no demonstration on compounds or real materials) and reproducibility concerns stemming from dependence on proprietary models like Gemini. The authors provide no plan to release computed embeddings or discuss computational costs, making it unclear how others could replicate or extend this work without access to the same commercial AI services.

**Questions:**

1. How do you handle elements with incomplete Wikipedia attribute coverage—do they receive fewer Local embeddings, default vectors, or some imputation method?
2. Did you compare Element2Vec against simpler baselines like linear regression on atomic features (e.g., atomic number, group, period) to quantify the advantage of text-derived embeddings?
3. How reliable was the LLM's sentence classification into attribute categories, and did misclassifications (especially for ambiguous sentences spanning multiple categories) impact embedding quality?
4. Why did concatenated Local embeddings underperform Global embeddings for regression, and did you explore learned fusion methods like attention mechanisms to weight attribute relevance for specific properties?
5. How sensitive is test-time training to hyperparameters and the ratio of known-to-unknown elements, and at what point does including too many unknowns cause overfitting or instability?
6. Is the Gemini embedding model publicly accessible, could alternative models like SBERT produce similar results, and will you release the Element2Vec embeddings for the 118 elements?

---

> ### Author Response · Authors · 2025-11-20
>
> > "The approach suffers from fundamental limitations in its dependence on Wikipedia as a single, uneven source of truth..." also mentioned in Question 1.
>
> For elements with incomplete Wikipedia attribute coverage, the corresponding local embeddings are simply omitted. However, the framework is fully flexible and can incorporate additional textual sources—such as ElementWiki, textbooks, database descriptions, or curated domain summaries, to complete missing attributes when available. We have also demonstrated that our workflow can process alternative text sources to generate embeddings, indicating that the approach is not restricted to Wikipedia and can readily integrate more structured or domain-specific text sources. We clarified this flexibility in the revised manuscript (See Figure. 11 in the SI: the performance of different text sources in prediction tasks)
>
> > "The paper provides no evidence that the LLM correctly categorizes sentences into attribute buckets or avoids hallucination during summarization...."
> For each element, we asked human experts in materials science to inspect the summaries produced by Gemini and evaluated whether the content was accurately derived from the source text, all 100 summaries were judged to be reasonably accurate and faithful to the original descriptions, with no cases of hallucination identified.
>
> > "The work's practical applicability is limited by its narrow scope (118 elements only, with no demonstration on compounds or real materials) and reproducibility concerns stemming from dependence on proprietary models like Gemini."
> Chemical elements are the **fundamental building blocks** of all materials, and establishing a robust, reliable representation at the elemental level is a necessary first step toward modeling chemical compounds and real materials. Our current work therefore focuses on encoding all 118 elements as a principled foundation. In future work, we will extend the framework to chemical compounds and bulk materials by composing elemental embeddings and incorporating structure-dependent information.
>
> > “Question 2: Did you compare Element2Vec against simpler baselines like linear regression on atomic features (e.g., atomic number, group, period) to quantify the advantage of text-derived embeddings?”
> We compared out results to the naive whole-text embeddings (Wikipedia2Vec: An efficient toolkit for learning and visualizing the embeddings of words and entities from Wikipedia). The Element2Vec workflow generated embeddings have much higher quality compared to the naive whole-text embedding models (see Figure. 12 in SI).
>
> >“Question 3: How reliable was the LLM's sentence classification into attribute categories, and did misclassifications (especially for ambiguous sentences spanning multiple categories) impact embedding quality?”
>
> Our workflow is designed to be able to handle the low-quality text resources. To verify the accuracy of Gemini's ability in classifying the sentences, we conducted a small-scale validation study to quantify the accuracy of the LLM in assigning sentences to our eight attribute categories. We posed 100 such classification questions to the Gemini embedding model and compared its outputs against the expert labels. The resulting accuracy provides an empirical estimate of the model’s reliability in capturing attribute semantics at the sentence level. Our evaluation shows that Gemini achieves strong performance across all four attribute categories, with scores of 0.93 (Mechanical), 0.96 (Thermal), 0.98 (Atomic and Radiational), and 0.99 (Chemical).
>
>
> > "Question 4: Why did concatenated Local embeddings underperform Global embeddings for regression, and did you explore learned fusion methods like attention mechanisms to weight attribute relevance for specific properties?"
>
> To show that this behavior in Figure 7(b) is not systematic, we conducted additional regression experiments on other scalar properties. We added Fig. 7(c) and 7(d) to report the normalized RMSE as a function of missing ratio for Young’s modulus and thermal conductivity, respectively. For Young’s modulus, the Local embeddings yield consistently lower RMSE than the Global embedding.
>
> > “Question 5: How sensitive is test-time training to hyperparameters ....?”
> We tested that TTT performance is not highly sensitive to its hyperparameters. We ran an additional experiment over learning rates and report the results in SI Table 4, suggesting that our TTT setup remains stable rather than overfitting.
> ﻿
> > “Question 6: Is the Gemini embedding model publicly accessible, could alternative models like SBERT produce similar results, and will you release the Element2Vec embeddings for the 118 elements?”
> Yes, Gemini embedding model publicly accessible. The raw textual content of many element pages is extremely long, the token length often exceeds the input limits of sentence‐embedding models such as SBERT. Gemini can process these oversized contexts without truncation.

---

> ### Author Response · Authors · 2025-11-25
> **Additional response to reviewer 9G3u**
>
> >"The authors provide no plan to release computed embeddings or discuss computational costs, making it unclear how others could replicate or extend this work without access to the same commercial AI services."
>
> The computed embeddings and key figures are publicly available at:
> https://github.com/2013261949/Element2Vec-Build-Chemical-Element-Representation-from-Text-for-Property-Prediction

---

> ### Author Response · Authors · 2025-11-27
>
> Dear Reviewer,
>
> I hope this message finds you well. As the discussion period is nearing its end with less than five days remaining, I wanted to ensure we have addressed all your concerns satisfactorily. If there are any additional points or feedback you’d like us to consider, please let us know. Your insights are invaluable to us, and we're eager to address any remaining issues to improve our work.
>
> Thank you for your time and effort in reviewing our paper.

---

### Meta-Review · Area_Chair_txeC · 2025-12-27

**Summary:**

The paper introduces a framework to compute vector representations for chemical elements (Fe, Br, etc.).

Reviewer had critical concerns that were not fully addressed during the rebuttal phase. Most importantly, there is a potential data leakage (wikipedia articles containing information about physical properties). All Reviewers criticized the lack of comparison to reasonable non-text baselines.

At this stage I am recommending rejection. Thank you for your submission and I hope the comments will be helpful in improving the manuscript.

**Reviewer Concerns:**

- Reviewer deg3 raised critical concerns that Wikipedia text often contains the exact target values
- Reviewers 9G3u, xsfF, EgL7, and deg3 all criticized the evaluation for lacking comparisons against standard non-text baselines used in materials science.

These critical concerns were not fully addressed during rebuttal.

**Reviewer Scores:**

The Reviewers would unlikely change their scores.

---

### Decision · Program_Chairs · 2026-01-26

Reject